# Gallium-Labeled PET Radiopharmaceuticals in Cardiovascular Disease

**DOI:** 10.3390/ph18030387

**Published:** 2025-03-09

**Authors:** Matthieu Bailly, Anne Claire Dupont, Guillaume Domain, Diane Darsin-Bettinger, Maxime Courtehoux, Gilles Metrard, Alain Manrique, Jonathan Vigne

**Affiliations:** 1Nuclear Medicine Department, CHU Orleans, 45100 Orléans, France; diane.darsin@chu-orleans.fr (D.D.-B.); gilles.metrard@chu-orleans.fr (G.M.); 2Laboratoire Interdisciplinaire pour l’Innovation et la Recherche en Santé d’Orléans, Orleans University, 45100 Orléans, France; 3Nuclear Medicine Department, CHU Tours, 37000 Tours, France; ac.dupont@chu-tours.fr (A.C.D.); m.courtehoux@chu-tours.fr (M.C.); 4Cardiology Department, CHU Orleans, 45100 Orléans, France; guillaume.domain@chu-orleans.fr; 5Nuclear Medicine Department, CHU Caen, 14000 Caen, France; manrique@cyceron.fr (A.M.); vigne-jo@chu-caen.fr (J.V.)

**Keywords:** gallium-labeled radiopharmaceuticals, cardiovascular, FAPI, DOTA, CXCR4, myocardial infarction, sarcoidosis, myocarditis

## Abstract

Gallium-labeled positron emission tomography (PET) probes targeting activated fibroblasts or somatostatin receptor expression are frequently used for varying applications in oncology. With the widespread availability of ^68^Ge/^68^Ga generators and cold kits, ^68^Ga tracers have become a main tool in molecular imaging. These tracers, such as [^68^Ga]Ga-DOTA-TATE, [^68^Ga]Ga-FAPI, and [^68^Ga]Ga-pentixafor, allow targeted imaging of the key pathological processes, including inflammation, fibrosis, and necrosis. This review highlights their potential in conditions like myocardial infarction, cardiac sarcoidosis, myocarditis, and other cardiomyopathies. Clinical and preclinical studies underscore their utility in visualizing active disease processes, predicting outcomes, and guiding therapeutic strategies. However, challenges remain, including the need for standardization, larger clinical trials, and integration into routine practice. These advancements position ^68^Ga-based PET as a promising modality for enhancing diagnostic precision and personalized treatment in cardiovascular disease.

## 1. Introduction

Positron emission tomography (PET) imaging has emerged as a crucial modality in the evaluation of cardiovascular diseases (CVDs), offering high sensitivity and specificity for detecting functional and molecular changes in cardiac tissues [1]. Radiopharmaceuticals are compounds that combine a radioactive isotope with a molecule that targets specific tissues, allowing for the detection and imaging of physiological processes at the molecular level. Among the various PET tracers, gallium-68 [^68^Ga]-labeled radiopharmaceuticals are gaining increasing attention due to their unique properties. Indeed, ^68^Ga (*t*_1/2_ = 67.71 min, β^+^ 89%, E_β+ max_ 1.9 MeV; EC 11%, E_γ max_ 4.0 MeV) is a radiometal that decays via positron emission and is well suited for PET imaging. Since the commercial availability of ^68^Ge/^68^Ga generators for clinical use with a shelf life of up to 12 months, it has become possible to collect daily [^68^Ga]GaCl_3_, which meets the European Pharmacopeia’s specifications [2]. This achievement fostered the introduction of radiopharmaceutical cold kits, allowing in-house production without requiring a cyclotron or highly qualified chemists. Consequently, ^68^Ga-based PET radiopharmaceuticals have become readily available in numerous nuclear medicine departments [3]. The usual design of ^68^Ga-labeled radiopharmaceuticals consists of a bifunctional chelator capable of retaining the radiometal bound to the molecule in the in vivo environment, and a ligand specific to a receptor biomarker. Generally, acyclic (HBED-CC) and macrocyclic polyaminocarboxylates (NOTA and DOTA) chelating agents have been shown to exhibit desirable properties, leading to the development of ^68^Ga-labeled somatostatin analogs (e.g., [^68^Ga]Ga-DOTA-TATE, [^68^Ga]Ga-DOTA-TOC), prostate-specific membrane antigen (PSMA) radioligands (e.g., [^68^Ga]Ga-PSMA-11), and fibroblast activation protein (FAP) inhibitors (e.g., [^68^Ga]Ga-FAPI-46 and [^68^Ga]Ga-FAPI-04) as commercial and potential PET imaging pharmaceuticals, respectively [4,5].

In comparison to other PET tracers, such as the widely used [^18^F]-FDG, [^68^Ga]-based tracers offer several notable advantages. [^18^F]-FDG is commonly used to assess glucose metabolism, particularly in oncology, but it has limitations in certain clinical scenarios, especially in evaluating tumors and CVDs, where receptor-targeted molecular imaging provides more specific and informative insights. In contrast, [^68^Ga]-based tracers, such as those targeting somatostatin receptors or PSMA, enable more targeted molecular imaging of specific receptor biomarkers, which are highly relevant in cardiovascular and oncological conditions. This specificity allows for a more detailed evaluation of disease processes at the molecular level, facilitating early diagnosis and more precise treatment monitoring. Furthermore, [^68^Ga]-based tracers offer significant theranostic potential, an important advantage over [^18^F]-FDG. Theranostics refers to the combined use of diagnostic and therapeutic agents, enabling both the detection of disease and the delivery of targeted therapy. For example, radiopharmaceuticals such as [^68^Ga]-PSMA-11, used in prostate cancer, can not only locate tumors but also guide targeted therapies, such as radiotherapy, making the ^68^Ga tracers a key tool for personalized medicine. This theranostic capability is not available with [^18^F]-FDG, which primarily serves as a diagnostic tool without the ability to directly guide therapy.

Gallium-labeled PET tracers have shown potential in targeting the key pathological processes in CVDs, including inflammation and fibrosis (Figure 1) [6]. For instance, gallium-labeled somatostatin analogs, initially developed for oncological purposes, have demonstrated significant utility in cardiac sarcoidosis and myocarditis imaging. The mechanism underlying the use of somatostatin analogs relies on their high affinity for somatostatin receptors (SSTRs), particularly subtype 2 (SSTR2), which are overexpressed by activated macrophages and inflammatory cells enabling the precise localization of inflammatory lesions and provide valuable information on disease activity. Similarly, FAP inhibitors target fibroblast activation protein, which is selectively expressed by activated fibroblasts during tissue remodeling and fibrosis. This mechanism makes gallium-labeled FAP inhibitors particularly useful for imaging fibrotic processes in post-infarction remodeling and chronic heart disease [7]. In addition to these tracers, [^68^Ga]Ga-pentixafor, a radiopharmaceutical targeting the C-X-C chemokine receptor type 4 (CXCR4), has emerged as a promising tool for imaging inflammatory and ischemic processes in cardiovascular diseases. CXCR4 plays a critical role in myocardial repair following ischemia by mediating the recruitment of inflammatory cells and stem cells to the injured myocardium. Moreover, its use in detecting vulnerable plaques in coronary artery disease represents a novel approach for assessing atherosclerotic plaque activity and stability [8].

Despite promising preliminary data, the clinical use of gallium-labeled radiopharmaceuticals in cardiovascular imaging is still in its early stages, with a need for preclinical and clinical data to validate their utility [9,10]. This review aims to provide a comprehensive analysis of gallium-labeled PET radiopharmaceuticals in CVDs, focusing on their applications in coronary artery disease (CAD), myocardial infarction, sarcoidosis, myocarditis, and other cardiomyopathies.

## 2. Coronary Artery Disease and Myocardial Infarction

After myocardial infarction, left ventricular remodeling is an important prognostic factor that can lead to heart failure [11]. Activated cardiac fibroblasts have been identified as the key mediators of the reparative response [12]. In a longitudinal rat model, [^68^Ga]Ga-FAPI04 uptake peaked in the injured area at day 6 after the acute infarction, and then gradually decreased, returning to near the baseline by 2 weeks [13]. The authors also performed an ex vivo analysis using immunofluorescence staining and demonstrated the presence of FAP-avid myofibroblasts in the infarcted myocardium. [^68^Ga]Ga-FAPI-04 accumulated mainly at the border zone of the infarcted myocardium, up to 3-fold higher than in the infarct zone. These results have been confirmed in another animal study [14].

Diekmann et al. conducted a prospective study on 35 patients after acute myocardial infarction, including Cardiac Magnetic Resonance (CMR), perfusion SPECT and [^68^Ga]Ga-FAPI-46 at day 11 [15]. The area of FAP upregulation was significantly larger than the SPECT perfusion defect size, and larger than late gadolinium enhancement or edema on CMR. FAPI uptake appeared to be a predictive factor of the evolution of ventricular dysfunction, with a significant inverse correlation with the left ventricle ejection fraction obtained at later follow-up (*r* = −0.58, *p* = 0.007). Kessler et al. confirmed consistent intense FAPI uptake within both the infarct and neighboring border zone, and a very good agreement with the affected coronary territory [16]. Xie et al. and Zhang et al. also confirmed the predictive value of FAPI uptake for predicting left ventricle remodeling [17,18]. Some case reports have also illustrated FAPI uptake greater than the infarct size [19]. [^68^Ga]Ga-FAPI-46 uptake has been observed to persist in late imaging in retrospective studies, with PET/MR performed at 30 days after infarction, and even at 16 months in a single patient [20]. An elevated FAP signal could be an early fibroblast response after reperfusion and its immediate inflammatory reaction. It might precede extracellular matrix abnormalities seen on CMR that establishes overt fibrosis. However, future research is needed to support this hypothesis.

[^68^Ga]Ga-Pentixafor PET imaging targets CXCR4, a receptor implicated in inflammation-related cardiovascular diseases, including progressive atherosclerosis and acute myocardial infarction. Indeed, CXCR4 is a transmembrane G-protein-coupled chemokine receptor that plays a crucial role in regulating cell movement across the body, particularly in immune cell trafficking. This makes it significant in inflammatory and autoimmune diseases, as well as in cancer metastasis, where it is being explored as a therapeutic target. Recently, CXCR4 has gained attention in cardiovascular disease, especially in conditions like atherosclerosis and acute myocardial infarction, where it, along with its ligand, CXCL12, contributes to leukocyte recruitment to injured areas, driving the inflammatory response. Additionally, CXCR4 is involved in angiogenesis and regulates the homing, mobilization, and survival of progenitor cells, linking it to myocardial ischemia and injury-induced restenosis. While its role in these processes is clearer, the significance of CXCR4 in native atherosclerosis remains uncertain. CXCR4 has also been reported to mediate leukocyte chemotaxis in specific inflammatory diseases, and a similar role in inflammatory cell recruitment has been suggested in myocardial ischemia. However, the importance of CXCR4-induced leukocyte recruitment to atherosclerotic lesions in vivo requires further investigation [21]. Preclinical studies demonstrated a consistent signal strength and time course for 68Ga-Pentixafor imaging, though the clinical outcomes after acute myocardial infarction are more variable due to the complexities of patient conditions [22]. A preclinical PET study identified early CXCR4 upregulation predicting acute rupture and chronic contractile dysfunction [23]. Imaging-guided CXCR4 inhibition accelerated inflammatory resolution and improved outcome, supporting a molecular imaging-based theranostic approach to guide therapy after myocardial infarction. A study by Werner et al. highlighted the translational and clinical potential of CXCR4-targeted PET using the CXCR4-ligand [^68^Ga]Ga-pentixafor as a biomarker for early post-infarct myocardial inflammation. This imaging approach revealed varying degrees of inflammation, with elevated early signals correlating with a higher risk of adverse outcomes, underscoring its incremental prognostic value [24]. Building on ex vivo evidence of CXCR4 presence in atherosclerotic tissue, Weiberg et al. suggested that [^68^Ga]Ga-pentixafor PET/CT imaging could provide a specific and effective means of visualizing CXCR4 expression in the walls of large arteries. The uptake of [^68^Ga]Ga-pentixafor appears to be significantly associated with the burden of coronary plaques and various cardiovascular risk factors, hinting at its possible utility in identifying vulnerable plaques. These results raise the possibility that this imaging modality could play a role in improving the detection and risk assessment of cardiovascular conditions [25].

SSTR PET/CT could also be helpful for detecting inflammatory lesions after myocardial infarction, as reported in a sub-study of six patients [26] or in a case report of silent myocardial infarction in a patient with pheochromocytoma [27].

Recently, a study investigated the use of [^68^Ga]Ga-NOTA-anti-MMR Nb for imaging macrophages during the reparative phase following myocardial infarction. By targeting the mannose receptor (MR), a marker for anti-inflammatory macrophages, the study aimed to evaluate the role of these cells in scar formation and tissue healing. In murine models, PET/CT imaging showed significant tracer uptake in the infarcted myocardium, which was confirmed through ex vivo analyses and histology. The results suggest that [^68^Ga]Ga-NOTA-anti-MMR Nb could be a valuable tool for visualizing the macrophages involved in post-MI wound healing and assessing therapeutic strategies to modulate inflammation [28].

The peptide, DOTA-ECL1i (1,4,7,10-tetraazacyclodo-decane-1,4,7,10-tetraacetic acid)-(extracellular loop 1 inverso), can bind to the chemokine (C–C motif) receptor, CCR2. Studies in animal post-myocardial infarction have shown transient increased [68Ga]Ga-DOTA-ECL1i uptake in injured myocardium, correlating with CCR2-positive macrophage abundance, and inversely correlating with contractile function 28 days post-ischemia-reperfusion injury [29]. These previous data have not been confirmed in human studies.

## 3. Sarcoidosis

Sarcoidosis is a multisystem granulomatous disorder of unknown etiology that can virtually involve all organ systems, including the heart. In a series of autopsied patients, myocardial granulomas were found in one-third of the patients, often clinically silent [30]. However, cardiac involvement can cause sudden death and requires early diagnosis and treatment. Diagnosis remains challenging, with a poor sensitivity of endomyocardial biopsy (20–30% due to a patchy, multifocal pattern of the disease) [31]. Unexplained Auriculo-Ventricular block in young patients or sustained monomorphic ventricular tachycardia of unknown etiology should lead to imaging [32]. CMR and [^18^F]F-fluorodeoxyglucose (FDG) PET/CT have proven their value for cardiac sarcoidosis detection and prognostication [33,34,35]. Indeed, FDG is avidly taken up by the activated macrophages, epithelioid cells, and Langerhans giant cells found in sarcoid granulomas. However, the specificity of [^18^F]F-FDG PET relies on the extinction of physiological uptake, requiring dedicated patient preparation [36,37,38]. Unlike normal cardiomyocytes, activated inflammatory cells (epithelioid cells, multinucleated giant cells, and some macrophages that are typically found in sarcoid granulomas) express SSTRs on their surface, particularly SSTR2A [39]. The feasibility of the somatostatin receptor targeted imaging of inflammatory cells had been previously demonstrated with ^111^In-labeled pentetreotide; however, due to SPECT’s lower spatial resolution, cardiac involvement has not been described [40]. Thanks to better resolution, PET targeting the SSTR has been demonstrated as a more specific alternative to FDG for imaging cardiac sarcoidosis [41].

A prospective study on nineteen patients, with three patients fulfilling the criteria for cardiac sarcoidosis, compared the results of [^18^F]F-FDG PET/CT and [^68^Ga]Ga-DOTANOC PET/CT, with an excellent diagnostic accuracy: 79% for [18F]F-FDG PET/CT and 100% [68Ga]Ga-DOTANOC PET/CT [42]. However, these results were derived from a very small sample. Even though they are encouraging, it is important to remember that ^1^⁸F-FDG PET/CT remains an effective diagnostic tool, with a sensitivity of 89% and a specificity of 78% [34]. [^68^Ga]Ga-DOTANOC seemed to be helpful because of an important proportion of [^18^F]F-FDG PET being inconclusive, due to poor cardiac preparation. In another prospective study on 15 patients with proven sarcoidosis and a suspicion of cardiac involvement, [^68^Ga]Ga-DOTATOC PET/CT was also concordant with CMR (96.1% on the segmental analysis), showing a high SUVmax and mean in inflamed areas [43]. However, a more recent study found poorer agreement between CMR and [^68^Ga]Ga-DOTANOC PET/CT on 17 patients (76.5%) [44]. The authors reported four discordant patients, with CMR positive and normal PET; they advanced the hypothesis that late gadolinium enhancement on CMR does not specifically differentiate between inflammation and fibrosis. SSTR PET might be better than CMR in identifying patients with active inflammation.

Some case reports also highlight the ability of SSTR PET/CT to monitor the therapeutic response of cardiac sarcoidosis [45,46]. [^68^Ga]Ga-FAPI PET/CT could also be helpful, to monitor cardiac fibroblast activity and identify ongoing cardiac remodeling [47,48,49].

## 4. Myocarditis

Myocarditis is an inflammatory cardiac disorder, predominantly induced by viruses [50], or less likely by a wide variety of toxic substances and drugs (such as immune checkpoint inhibitors) [51] and systemic immune-mediated diseases [52]. It remains of poor prognosis because of the progression to left ventricular dysfunction, heart failure, or arrhythmias which are associated with a poor prognosis.

CMR imaging is considered the non-invasive gold-standard method for diagnosing myocarditis [53,54], but remains challenging in certain patients due to irregular cardiac rhythms, claustrophobia, cardiac implantable electronic devices, or severe obesity. In addition, it lacks the sensitivity to detect subacute or chronic myocarditis and cannot establish the type of myocarditis (i.e., specific infiltration of immune cells and underlying etiology). [50]. A low sensitivity of CMR has also been reported in immune checkpoint inhibitor-related myocarditis in a series of 103 patients [55].

FDG PET could be used to highlight inflammation, such as in cardiac sarcoidosis. However, the diagnosis of myocarditis remains challenging. SSTR PET imaging may be an alternative as SSTRs are minimally expressed by cardiac cells under normal physiological conditions. Indeed, no significant cardiac uptake is reported in SSTR PET scans performed in oncological indications [56]. However, SSTRs are highly expressed on lymphocytes, macrophages, and activated monocytes—the primary cell types involved in myocarditis [57].

Some case reports have highlighted the potential of [^68^Ga]Ga-DOTATOC ECG-triggered digital PET to delineate areas of myocarditis following COVID-19 vaccination [58].

A pilot study on six patients with active peri-/myocarditis showed a similar pattern of SSTR expression in PET/CT and abnormal signal in CMR, with an overall concordance of 85.3% on segmental analysis, suggesting a close spatial resolution of macrophage concentration and structural changes [26]. A recent preliminary study confirmed these preliminary findings on 14 patients with acute myocarditis via CMR. [^68^Ga]Ga-DOTATOC PET/CT was performed within 72 h and 4 months later in 10 patients [59]. The authors reported a myocardial/blood SUVmax ratio of 2.18 being a diagnosis threshold, in focal, multifocal, and diffuse patterns. The SUVmax ratio decreased at 4 months, but there was still some 68Ga-DOTATOC uptake. It is important to note that the authors planned to include a total of 30 patients. The final results will confirm these observations.

Another study included 11 patients with suspected immune checkpoint inhibitor-related myocarditis [60]. Nine of them underwent [^68^Ga]Ga-DOTATOC PET/CT with a high myocardial/blood SUVpeak ratio (mean 3.2 ± 0.8, range 2.2–4.4); among them, eight had concomitant CMR, with three having lesions evocative of myocarditis. Some patients also showed signs of associated myositis. The authors concluded that SSTR PET might be a sensitive tool for immune checkpoint inhibitor-related myocarditis along with newly identified immune correlates.

A recent retrospective study of 511 oncological [^68^Ga]Ga-DOTATOC PET/CT showed significant myocardial uptake in 26.9% [61]. Among them, 31 patients were recruited at the time of acute myocarditis diagnosis. An increased uptake was consistently seen on at least two contiguous left ventricle segments, associated with a marked decrease in blood activity, and with myocardial uptake predominantly localizing to inferior and inferior-lateral segments. The myocardial/blood SUVmax ratio threshold of 2.2 yielded a sensitivity of 87% and specificity of 77.4% for the diagnosis of acute myocarditis compared to the less intense and less extensive uptake described in the other patients. Adding the uptake volume could enhance the diagnostic performance rates. These preliminary findings suggest that SSTR PET/CT could be an additional tool to CMR when the diagnosis of myocarditis is challenging (Figure 2).

Less data have been reported with FAPI PET/CT. A series of three cases of myocarditis of varying duration (7 h, 1 week, 1 month) have reported different [^18^F]F-FAPI uptakes, helping to evaluate the extent of fibrosis caused by myocarditis [62].

Other targets might be of interest: the arginine-glycine-aspartic acid (RGD) peptide expressed by αvβ3-integrin in macrophages ([^68^Ga]-Ga-NODAGA-RGD) [63] or mannose receptor expression by macrophages ([^68^Ga]-Ga-NOTA-MSA) [64].

## 5. Other Cardiomyopathies

Amyloidosis is a systemic multiorgan infiltrative disease characterized by the extracellular deposition of insoluble proteins. The disease has two main subtypes, transthyretin cardiac amyloidosis (ATTR-CA) and immunoglobulin light chain cardiac amyloidosis (AL-CA), characterized by the nature of the infiltrating protein [65]. CA is widely underdiagnosed and increasingly recognized as a cause of heart failure with a preserved ejection fraction [66]. Amyloid deposition can lead to cardiomyocyte necrosis and interstitial fibrosis [67]; thus, the 68Ga-labeled fibroblast activation protein inhibitor could help to identify and characterize activated fibroblasts in cardiac amyloidosis. [^68^Ga]Ga-FAPI PET uptake has been described in two cases: one of ATTR cardiac amyloidosis [68] and one of AL [69]. A first prospective study included 30 patients with proven diagnoses of AL amyloidosis on endomyocardial biopsy or extracardiac tissue, 27 of them having cardiac involvement [70]. [^68^Ga]Ga-FAPI-04 PET showed increased left ventricle uptake in 88.9% of patients with cardiac amyloidosis, mostly in an extensive pattern. There was also a good correlation of FAPI uptake with the NT-proBNP level, and other imaging such as echocardiography and CMR (septal thickness, ejection fraction, left ventricle end-systolic volume, global circumferential and longitudinal strain, extracellular volume). The same team recently added another prospective study to determine the molecular phenotypes of AL amyloidosis using 18F-florbetapir and [^68^Ga]Ga-FAPI-04 PET/CT imaging to assess myocardial amyloid deposition and fibroblast activation. The study highlights that a higher uptake on [^68^Ga]Ga-FAPI-04 PET/CT imaging may correlate with worse clinical outcomes in newly diagnosed AL-CA, suggesting that this metric could serve as a molecular imaging tool complementary to [^18^F]F-florbetapir imaging. These findings suggest that dual-tracer PET/CT could be a valuable tool for understanding the amyloid dynamics and advancing fibrosis prevention [71].

Myocardial fibrosis is a non-specific process in all heart muscle disease. Thus, FAPI imaging could be useful in many applications [72]. [^68^Ga]Ga-FAPI uptake has been reported in a preclinical and in a preliminary clinical study in patients with heart failure [73]. FAPI uptake and ventricular wall motion decreased over time as cardiac fibrosis and the degree of myocardial injury gradually increased. [^18^F]AlF-NOTA-FAPI-04 PET/CT has been reported as a promising and non-invasive method to assess the progression of fibrosis in heart failure with a preserved ejection fraction to facilitate the clinical management [74].

Increased myocardial [^68^Ga]Ga-FAPI uptake has been reported in a young male patient with dilated cardiomyopathy [75], and in a male patient with hypertensive heart disease [76]. A recent study highlighted the added value of FAPI PET for diagnosis and risk assessment in patients with hypertrophic cardiomyopathy [77]. FAPI uptake could reflect the potential strain reduction seen in CMR [78] and was associated with a 5-year risk of sudden cardiac death in hypertrophic cardiomyopathy [79]. As myocardial fibrosis is an important stratification factor for the occurrence of ventricular arrhythmias in hypertrophic obstructive cardiomyopathy, FAPI uptake could be useful for guiding surgical strategies [80]. These observations were confirmed in a pilot study including patients with non-ischemic cardiomyopathies and [^68^Ga]Ga-FAPI-04 PET/CT [81].

Up to now, right ventricle evaluation remains challenging with CMR. [^68^Ga]Ga-FAPI uptake has been described in two case reports and of patients with chronic thromboembolic pulmonary hypertension [82,83]. Two prospective studies confirmed FAPI uptake in the right ventricular free wall which correlated positively with the wall thickness and negatively with the right ventricular function [84,85].

In onco-cardiology, [^68^Ga]Ga-FAPI left ventricle uptake was first described in a patient with chemotherapy-associated cardiotoxicity [86]. In an analysis of 229 patients, those treated with anthracyclines or alkylating agents had an unexpectedly high myocardial [^68^Ga]Ga-FAPI uptake [87]. In a rat model, [^68^Ga]Ga-FAPI PET/CT can be used for the early detection of active myocardial fibrosis in anthracycline-induced cardiotoxicity and the evaluation of the efficacy of therapeutic interventions [88]. There was also an association between [^68^Ga]Ga-FAPI high uptake and previous radiotherapy [87]. This association between radiation and myocardial damage has also been reported in a study including animals and patients [89].

## 6. Conclusions

[^68^Ga]Ga-labeled PET radiopharmaceuticals have ushered in a new era of precision imaging in cardiovascular diseases, enabling the visualization of inflammation, fibrosis, and ischemia with unprecedented specificity. Their utility has been demonstrated across diverse pathologies, from acute myocardial infarction to chronic cardiomyopathies and are summarized in Table 1. Despite encouraging preclinical and clinical data, the transition to routine clinical application is hindered by limited large-scale studies and the standardization of protocols. Future research should aim to validate these findings in larger cohorts and explore the theranostic potential of [^68^Ga]Ga-based tracers. Integrating these innovations into clinical practice has the potential to revolutionize CVD diagnosis, risk stratification, and therapeutic monitoring.

Several challenges remain. One of the main limitations is the availability and accessibility of gallium-68, despite the development of generators and cyclotron-based production. Additionally, its short half-life (68 min) restricts its use to well-equipped centers and poses logistical challenges for radiopharmaceutical production and distribution. The use of certain fluorinated radiopharmaceuticals, such as [^1^⁸F]F-FAPI, could help overcome these limitations. However, PET accessibility remains limited for cardiovascular studies due to the high demand for oncological examinations. Finally, it remains to be seen whether these new tracers have the potential to enhance our understanding of the pathophysiology of processes involved in cardiac remodeling and, consequently, influence patient management in one way or another.

## Figures and Tables

**Figure 1 pharmaceuticals-18-00387-f001:**
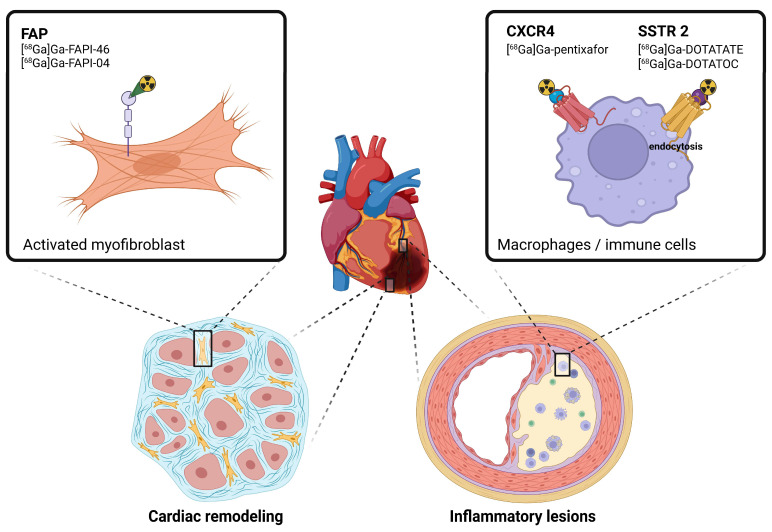
Schematic representation of gallium-68-labeled radiopharmaceuticals and their targets in the context of cardiovascular diseases. [^68^Ga]Ga-DOTATATE and [^68^Ga]Ga-DOTATOC accumulates in macrophages and inflammatory cells via endocytosis after binding to the somatostatin receptor type 2 (SSTR2). Molecular imaging of vascular inflammation can also be monitored by targeting C-X-C chemokine receptor type 4 (CXCR4) using [^68^ Ga]Ga-pentixafor. Cardiac remodeling following myocardial infarction implicates activated myofibroblasts that overexpress fibroblast activation protein (FAP).

**Figure 2 pharmaceuticals-18-00387-f002:**
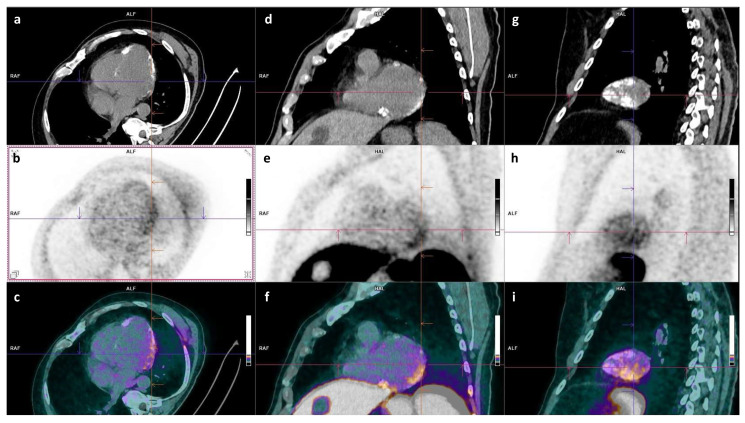
Example of calcifying myocarditis under Revlimid on [^68^Ga]-GaDOTATOC PET/CT ((**a**–**c**): short-axis slices, (**d**–**i**): long-axis slices). A 66-year-old patient showing increased left myocardial uptake, particularly in the infero-latero-basal region, adjacent to subepicardial calcifications (myocardial/blood SUVmax ratio of 2.75). CMR was inconclusive due to the extent of calcifications.

**Table 1 pharmaceuticals-18-00387-t001:** Summary of Gallium-labeled PET radiopharmaceuticals’ potential cardiovascular applications.

Targets		Somatostatin Receptors SSTR	Fibroblast Activation Protein FAP	Chemokine Receptor Type 4 CXCR4
Radiopharmaceuticals		[68Ga]Ga-DOTA-TATE [68Ga]Ga-DOTA-TOC	[68Ga]Ga-FAPI-46 [68Ga]Ga-FAPI-04	[^68^Ga]Ga-pentixafor
Type of cells involved		Macrophages, lymphocytes, monocytes	Activated fibroblasts	Inflammatory cells and stem cells
Applications	Ischemia Infarction	Potential use in inflammatory lesions after myocardial infarction (case series [26,27])	Preclinical: accumulation mainly at infarct border [13,14] Clinical: uptake larger than infarct size [15,16], predictive value of FAPI uptake for predicting left ventricle remodeling [17,18]	Potential biomarker for early post-infarct myocardial inflammation and potential prognostic value [23,24]
	Sarcoidosis	Excellent diagnostic accuracy [42,43] Some discordance with CMR [44]: active inflammation with PET vs. inflammation or fibrosis on CMR? Response assessment (case reports) [45,46]	Monitor cardiac fibroblast activity and identify ongoing cardiac remodeling (case reports) [47,48,49]	
	Myocarditis	Good correlation with CMR [26,59,60,61] Suggested myocardial/blood SUVmax ratio of 2.2	3 case reports with 18F-FAPI [62]	
	Amyloidosis		Good correlation with NT-proBNP, echocardiography, CMR in AL amyloidosis [70] Case reports (ATTR cardiac amyloidosis [68] and one of AL [69])	
	Others		Dilated cardiomyopathy (case report [75]) Hypertrophic cardiomyopathy [77,78,79] Chronic thromboembolic pulmonary hypertension: right ventricle uptake [82,83,84,85] Chemotherapy-associated cardiotoxicity [87,88]	

## Data Availability

No new data were created or analyzed in this study. Data sharing is not applicable to this article.

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
