# Peer review of "Gallium-Labeled PET Radiopharmaceuticals in Cardiovascular Disease"

_pharmaceuticals, 2025, doi:10.3390/ph18030387_

Round 1
Reviewer 1 Report
Comments and Suggestions for Authors
Comment
Positron Emission Tomography (PET) imaging shows great potential for precise diagnosis in cardiovascular diseases. The review paper by Bailly conclude the usage of Gallium-68 (⁶⁸Ga) labeled PET tracers in oncology and cardiovascular diseases. These probes show potential in conditions like myocardial infarction, cardiac sarcoidosis, myocarditis, and other cardiomyopathies, which help visualize active disease processes, predict outcomes, and guide therapeutic strategies. I would suggest the publication of the manuscript in Pharmaceuticals after addressing the following comments:
1 At the conclusion part, please give a more detailed description of the challenge and perspective of the gallium-labelled radiotracers.
2 As a review paper in PET radiopharmaceuticals, please add at least 1 figure in different parts (Except Abstract, Introduction and Conclusions) and give detailed description on it.
3 At line 100, gadolinium enhancement imaging was introduced but have no relation to gallium-labelled PET tracers. My suggestion is to delete it or replace the example with other PET tracers to detect focal. If not, I just can’t understand why you mention MRI. You need to add an explanation to make sure it seems logical.
4 Please keep the font of the Reference consistent with the previous manuscript.
Author Response
Comments 1: "At the conclusion part, please give a more detailed description of the challenge and perspective of the gallium-labelled radiotracers."
Response 1: Thank you for your insightful comment. We have expanded the conclusion to provide a more detailed discussion of the challenges associated with gallium-labelled radiotracers, including issues related to their production, and clinical applications : "Several challenges remain. One of the main limitations is the availability and accessibility of gallium-68, despite the development of generators and cyclotron-based production. Additionally, its short half-life (68 minutes) restricts its use to well-equipped centers and poses logistical challenges for radiopharmaceutical production and distribution. The use of certain fluorinated radiopharmaceuticals, such as [¹⁸F]F-FAPI, could help overcome these limitations. However, PET accessibility remains limited for cardiovascular studies due to the high demand for oncological examinations. Finally, it remains to be seen whether these new tracers have the potential to enhance our understanding of the pathophysiology of processes involved in cardiac remodeling and, consequently, influence patient management in one way or another."
Comments 2: "As a review paper in PET radiopharmaceuticals, please add at least 1 figure in different parts (Except Abstract, Introduction and Conclusions) and give detailed description on it."
Response 2: Thank you for pointing this out. We do not have imaging available in each section and we did not want to provide previously published images. However, to illustrate this review we included a myocarditis image using 68Ga-DOTATOC in the myocarditis section (added in red in the manuscript).
Comment 3: "At line 100, gadolinium enhancement imaging was introduced but have no relation to gallium-labelled PET tracers. My suggestion is to delete it or replace the example with other PET tracers to detect focal. If not, I just can’t understand why you mention MRI. You need to add an explanation to make sure it seems logical."
Response 3: Thank you for this comment. We chose to delete it to clarify the reading. Modification was done in the manuscript.
Comments 4: "4 Please keep the font of the Reference consistent with the previous manuscript."
Response 4: This has been modified.
Reviewer 2 Report
Comments and Suggestions for Authors
- The introduction lacks explanation of some aspects that the reader could be unfamiliar with. For example, explanation of why 68Ga-based tracers are particularly advantageous compared to other PET tracers such as 18F-FDG should be added.
- In the section on CXCR4-targeted imaging (108-128), the authors didn't elaborate about the molecular and mechanistic pathways of CXCR4 related inflammation and injury and "jumped" to its role in various CVDs. Expanding mechanistic aspects CXCR4-targeted imaging is more appropriate.
- Lines (160-173) the authors mention that SSTR CT/PET is superior to FDG CT/PT in sarcoidosis diagnosis. However, the authors didn't introduce any numerical confirmation for this superiority in terms of specificity or sensitivity. Moreover, FDG CT/PET is underestimated by the authors although it is considered an effective tool in diagnosis with a sensitivity of 89% and speificity of 78% according to Youssef et al. (2012) (DOI: https://doi.org/10.2967/jnumed.111.090662)
- The authors should highlight the advantages of FDG CT/PET and provide numerical confirmation for their assumption of superiority of SSTR CT/PET over FDG CT/PT.
- In myocarditis section (186-207) the authors mention that SSTR PET could replace CMR in certain scenarios. However, CMR remains the gold standard, and PET is complementary rather than a replacement. This should be clarified.
- The phrase “challenges remain, including the need for standardization, larger clinical trials, and integration into routine practice” provides little information on what standardization issues are present, or what type of clinical trials are needed
- The subsection Coronary Artery Disease and Myocardial Infarction concludes that “FAPI uptake appeared to be a predictive factor of the evolution of ventricular dysfunction” but does not state how this was ascertained or if any statistical analyses were performed
- The conclusion section briefly mentions the need for larger studies but does not elaborate on specific research gaps. Providing targeted recommendations for future research would strengthen the manuscript.
- The review predominantly highlights positive findings regarding 68Ga tracers but does not adequately discuss negative studies or potential pitfalls.
- The manuscript could benefit from additional figures, such as schematic illustrations of the mechanisms of action for different tracers
- Line 36: "decay by positron emission "should be "decays by positron emission."
- Line 97: "persist in late imaging in a retrospective study" should be revised for clarity, e.g., "has been observed to persist in late imaging in retrospective studies."
- Not all abbreviations are defined when first used (e.g., "SSTR" on line 57).
Author Response
Comments 1: "The introduction lacks explanation of some aspects that the reader could be unfamiliar with. For example, explanation of why 68Ga-based tracers are particularly advantageous compared to other PET tracers such as 18F-FDG should be added."
For clarity, we have included the definition of a radiopharmaceutical in the introduction. Additionally, a paragraph highlighting the advantages of 68Ga-PET tracers has been added as follows:
“In comparison to other PET tracers, such as the widely used [18F]-FDG, [68Ga]-based tracers offer several notable advantages. [18F]-FDG is commonly used to assess glucose metabolism, particularly in oncology, but it has limitations in certain clinical scenarios, especially in evaluating tumors and CVDs, where receptor-targeted molecular imaging provides more specific and informative insights. In contrast, [68Ga]-based tracers, such as those targeting somatostatin receptors or PSMA, enable more targeted molecular imaging of specific receptor biomarkers, which are highly relevant in cardiovascular and oncological conditions. This specificity allows for a more detailed evaluation of disease processes at the molecular level, facilitating early diagnosis and more precise treatment monitoring. Furthermore, [68Ga]-based tracers offer significant theranostic potential, an important advantage over [18F]-FDG. Theranostic refers to the combined use of diagnostic and therapeutic agents, enabling both the detection of disease and the delivery of targeted therapy. For example, radiopharmaceuticals such as [68Ga]-PSMA-11, used in prostate cancer, can not only locate tumors but also guide targeted therapies, such as radiotherapy, making the 68Ga tracers a key tool for personalized medicine. This theranostic capability is not available with [18F]-FDG, which primarily serves as a diagnostic tool without the ability to directly guide therapy.”
Comments 2: "In the section on CXCR4-targeted imaging (108-128), the authors didn't elaborate about the molecular and mechanistic pathways of CXCR4 related inflammation and injury and "jumped" to its role in various CVDs. Expanding mechanistic aspects CXCR4-targeted imaging is more appropriate."
Response 2: The molecular and mechanical pathways of inflammation and lesions related to CXCR4 have been developed as proposed.
“Indeed, CXCR4 is a transmembrane G-protein-coupled chemokine receptor that plays a crucial role in regulating cell movement across the body, particularly in immune cell trafficking. This makes it significant in inflammatory and autoimmune diseases, as well as in cancer metastasis, where it is being explored as a therapeutic target. Recently, CXCR4 has gained attention in cardiovascular disease, especially in conditions like atherosclerosis and acute myocardial infarction, where it, along with its ligand CXCL12, contributes to leukocyte recruitment to injured areas, driving the inflammatory response. Additionally, CXCR4 is involved in angiogenesis and regulates the homing, mobilization, and survival of progenitor cells, linking it to myocardial ischemia and injury-induced restenosis. While its role in these processes is clearer, the significance of CXCR4 in native atherosclerosis remains uncertain. CXCR4 has also been reported to mediate leukocyte chemotaxis in specific inflammatory diseases, and a similar role in inflammatory cell recruitment has been suggested in myocardial ischemia. However, the importance of CXCR4-induced leukocyte recruitment to atherosclerotic lesions in vivo requires further investigation.”
Comments 3: Lines (160-173) the authors mention that SSTR CT/PET is superior to FDG CT/PT in sarcoidosis diagnosis. However, the authors didn't introduce any numerical confirmation for this superiority in terms of specificity or sensitivity. Moreover, FDG CT/PET is underestimated by the authors although it is considered an effective tool in diagnosis with a sensitivity of 89% and speificity of 78% according to Youssef et al. (2012) (DOI: https://doi.org/10.2967/jnumed.111.090662)
The authors should highlight the advantages of FDG CT/PET and provide numerical confirmation for their assumption of superiority of SSTR CT/PET over FDG CT/PT.
Response 3: Thank you for this comment. We introduced the numerical findings directly in the manuscript. We also added a discussion about the small sample in the cited study, and the global effectiveness of FDG PET : "However, these results were derived from a very small sample. Even though they are encouraging, it is important to remember that ¹⁸F-FDG PET/CT remains an effective diagnostic tool, with a sensitivity of 89% and a specificity of 78%"
Comments 4: In myocarditis section (186-207) the authors mention that SSTR PET could replace CMR in certain scenarios. However, CMR remains the gold standard, and PET is complementary rather than a replacement. This should be clarified.
Response 4: This has been clarified in the manuscript "These preliminary findings suggest that SSTR PET/CT could be an additional tool to CMR when the diagnosis of myocarditis is challenging"
Comments 5: The phrase “challenges remain, including the need for standardization, larger clinical trials, and integration into routine practice” provides little information on what standardization issues are present, or what type of clinical trials are needed
Response 5: This phrase in the abstract remains. However, according to the other comments, we reviewed the manuscript by adding more data about the challenges and the need for standardization.
Comments 6: The subsection Coronary Artery Disease and Myocardial Infarction concludes that “FAPI uptake appeared to be a predictive factor of the evolution of ventricular dysfunction” but does not state how this was ascertained or if any statistical analyses were performed
Response 6: Thank you for this comment. We clarified by adding the numerical findings "(r = -0.58, P = 0.007)".
Comments 7: The conclusion section briefly mentions the need for larger studies but does not elaborate on specific research gaps. Providing targeted recommendations for future research would strengthen the manuscript. The review predominantly highlights positive findings regarding 68Ga tracers but does not adequately discuss negative studies or potential pitfalls.
Response 7: The conclusion has been expanded to better explain this point. We also added some limitations (small sample sizes, animal studies without human confirmation...)
Comments 8: The manuscript could benefit from additional figures, such as schematic illustrations of the mechanisms of action for different tracers
Response 8: Figure 1 has bee added.
Comments 9: Line 36: "decay by positron emission "should be "decays by positron emission."
Response 9: This has been replaced.
Comments 10: Line 97: "persist in late imaging in a retrospective study" should be revised for clarity, e.g., "has been observed to persist in late imaging in retrospective studies."
Response 10: This has been revised.
Comments 11: Not all abbreviations are defined when first used (e.g., "SSTR" on line 57).
Response 11: This abbreviation has been defined.
Reviewer 3 Report
Comments and Suggestions for Authors
Bailly et al. submitted a review manuscript titled " Gallium-labelled PET radiopharmaceuticals in cardiovascular disease.” The authors aimed to explore the potential application of 68Ga-based PET as a diagnostic and therapeutic tool for varied cardiovascular complications (coronary artery diseases, myocardial infarction, myocarditis, and cardiac sarcoidosis), besides their common use in oncology and inflammatory events. The introduction of the manuscript covered all the basic information about PET tracers focussing on gallium-68 [68Ga]-labelled radiopharmaceuticals. In nuclear medicines, it is obvious that radiopharmaceuticals can retain a radiometal bound to the molecule under in vivo conditions, leading to a ligand specific to a receptor/protein target. Since the actual clinical applications of 68Ga-labelled radiopharmaceuticals are not yet well-established, this review manuscript is more significant and interesting to researchers developing a novel therapeutic strategy in the management of cardiovascular complications. Authors highlighted the targeting CXCR4 receptor using [68Ga]Ga-Pentixafor PET, besides its use as a biomarker for early post-infarct myocardial inflammation. Similarly, authors presented recent clinical reports about the applications of such radiopharmaceuticals in other related CVS conditions including cardiomyopathies. The authors summarized their findings pertaining to Gallium-labeled PET radiopharmaceuticals for cardiovascular interventions in Table 1. The manuscript concluded that the a need for further research in larger cohorts to demonstrate the theranostic potential of [68Ga]Ga-based tracers. This manuscript is well organized and includes all the recent open-sourced literature reports. The authors also addressed current drawbacks and warrants to explore the practical applicability of Gallium-labelled PET radiopharmaceuticals. The following comments need to be addressed before the acceptance of the manuscript
- A figure needs to be given describing the mechanism of any one of the Gallium-labelled PET radiopharmaceuticals in the treatment of CVS diseases.
- Authors focussed on three target receptors (SSTR, FAP, and CXCR4) and need to describe other potential key proteins.
- All references need to be formatted and presented as per the guidelines of the journal
- Authors need to discuss any patents published related to the studied subject.
- Future perspectives can be included along with the conclusion.
Author Response
Comments 1: A figure needs to be given describing the mechanism of any one of the Gallium-labelled PET radiopharmaceuticals in the treatment of CVS diseases.
Response 1: Figure 1 has been added.
Comments 2: Authors focussed on three target receptors (SSTR, FAP, and CXCR4) and need to describe other potential key proteins.
Response 2: Thank you for this comment. We already discussed [68Ga]Ga-NOTA-anti-MMR Nb in myocardial infarction. We added also some insights about [68Ga]Ga-DOTA-ECL1i in myocardial infarction, [68Ga]-Ga-NODAGA-RGD and [68Ga]-Ga-NOTA-MSA in myocarditis.
Comments 3: All references need to be formatted and presented as per the guidelines of the journal
Response 3: The references have been formatted.
Comments 4: Authors need to discuss any patents published related to the studied subject.
Response 4: No patents published related to the studied subject need to be discussed.
Comments 5: Future perspectives can be included along with the conclusion.
Response 5: According to the first reviewer comment, we expanded the conclusion this way.
Round 2
Reviewer 2 Report
Comments and Suggestions for Authors
Thank you for taking all comments into consideration. The manuscript is better now.